# Coupled carbon and nitrogen cycling regulates the cnidarian–algal symbiosis

Nils Rädecker [1] ✉, Stéphane Escrig[1], Jorge E. Spangenberg [2], Christian R. Voolstra [3] & Anders Meibom [1,4]

Efficient nutrient recycling underpins the ecological success of cnidarian-algal symbioses in oligotrophic waters. In these symbioses, nitrogen limitation restricts the growth of algal endosymbionts *in hospite* and stimulates their release of photosynthates to the cnidarian host. However, the mechanisms controlling nitrogen availability and their role in symbiosis regulation remain poorly understood. Here, we studied the metabolic regulation of symbiotic nitrogen cycling in the sea anemone Aiptasia by experimentally altering labile carbon availability in a series of experiments. Combining $^{13}C$ and $^{15}N$ stable isotope labeling experiments with physiological analyses and NanoSIMS imaging, we show that the competition for environmental ammonium between the host and its algal symbionts is regulated by labile carbon availability. Light regimes optimal for algal photosynthesis increase carbon availability in the holobiont and stimulate nitrogen assimilation in the host metabolism. Consequently, algal symbiont densities are lowest under optimal environmental conditions and increase toward the lower and upper light tolerance limits of the symbiosis. This metabolic regulation promotes efficient carbon recycling in a stable symbiosis across a wide range of environmental conditions. Yet, the dependence on resource competition may favor parasitic interactions, explaining the instability of the cnidarian-algal symbiosis as environmental conditions in the Anthropocene shift towards its tolerance limits.

Photosymbioses between heterotrophic hosts and phototrophic symbionts are diverse and widespread in the aquatic environment[1]. The efficient recycling of organic and inorganic nutrients in these associations provides a critical advantage under oligotrophic conditions and has enabled their repeated evolutionary formation[2,3]. Coral reefs, with their immense biodiversity and productivity, are testimony to this ecological success and the fundamental role of photosymbioses in the marine environment[4,5]. Yet, the dependence of these ecosystems on the cnidarian-algal symbiosis may also prove their Achilles heel in times of global change. Marine heatwaves, among other environmental disturbances, now repeatedly cause the disruption of this symbiosis in so-called mass bleaching events that result in widespread reef degradation[6]. Thus, understanding the processes that maintain the stable cnidarian-algal symbiosis could elucidate the evolutionary origins of photosymbioses and help understand their apparent susceptibility to the accelerating environmental change of the Anthropocene.

The photosynthetic activity of algal symbionts implies that the functioning of the cnidarian-algal symbiosis is intimately linked to light availability[7]. Latitudinal, seasonal, and tidal fluctuations in light intensity, attenuation with depth, shading, and turbidity differences create a complex mosaic of light conditions in aquatic environments[8–10]. Such

[1]Laboratory for Biological Geochemistry, School of Architecture, Civil and Environmental Engineering, École Polytechnique Fédérale de Lausanne (EPFL), Lausanne, Switzerland. [2]Institute of Earth Surface Dynamics, University of Lausanne, Lausanne, Switzerland. [3]Department of Biology, University of Konstanz, Konstanz, Germany. [4]Center for Advanced Surface Analysis, Institute of Earth Sciences, University of Lausanne, Lausanne, Switzerland. ✉e-mail: nils.radecker@epfl.ch

variability poses a challenge to photosynthetic organisms as low light levels may limit photosynthetic carbon fixation, while high light levels may result in excessive photooxidative damage[7,11]. Yet, cnidarian-algal symbioses can be found across a wide range of light regimes ranging from intertidal to mesophotic environments[12,13]. The key to this broad ecological tolerance lies in the efficient photo-acclimation of both symbiotic partners. Specifically, changes in symbiont densities, host pigments, morphology, and behavior (e.g., locomotion) may modulate light microenvironments for algal symbionts within the host[14–18]. Likewise, changes in photosynthetic pigments and antioxidant levels of the algae enable optimal light harvesting while avoiding excessive photodamage[18–20]. However, holobiont responses to changes in light availability are highly species- and context-dependent and are often confounded by changes in other environmental parameters (e.g., along depths gradients)[15,21]. Our understanding of the general regulatory processes shaping the ecological niche of cnidarian-algal symbioses is thus limited.

The regulation of the cnidarian-algal symbiosis is directly linked to the nutrient exchange between the host and its symbionts[22]. In this symbiosis, the release of excess photosynthates by the algae supports the metabolic energy demands of the host[16]. Host respiration enhances $CO_2$ availability for algal photosynthesis, forming an efficient recycling loop that supports high gross productivity[23]. Constant nutrient limitation is required to initiate and maintain carbon translocation and recycling in the symbiosis[24,25]. In a stable state, low nitrogen availability limits algal symbiont growth and ensures the translocation of excess carbon to the host[26–29]. Consequently, the onset of nitrogen limitation has been proposed to play a vital role in the establishment of the cnidarian-algal symbiosis as it promotes life stage transition and controls the population density of the algal endosymbionts[30–32]. Likewise, failure to maintain nitrogen-limited conditions has been linked to the breakdown of the symbiosis during heat stress, i.e., bleaching[33–35].

However, while the importance of nitrogen cycling in the cnidarian-algal symbiosis is widely accepted, the factors regulating nitrogen availability *in hospite* remain poorly understood[36,37].

Previous studies propose that algal symbiont nitrogen limitation arises from symbiont-symbiont as well as host-symbiont competition for inorganic nitrogen[30,34,38–43]. Deciphering the factors regulating this interplay of intra- and interspecific nitrogen competition could be key to understanding the functional regulation of the symbiosis. Here, we thus investigated the nutritional and environmental controls of nutrient cycling in the cnidarian-algal symbiosis to elucidate the processes shaping its ecological niche and tolerance limits. Using the photosymbiotic model organism Aiptasia (*Exaiptasia diaphana* clonal line CC7 harboring *Symbiodinium linucheae* strain SSA01 endosymbionts)[44,45], we conducted two complementary experiments to elucidate the role of nitrogen competition in shaping symbiotic interactions. First, we investigated the effects of labile carbon availability on host ammonium assimilation and its consequences on nitrogen availability for algal symbionts. Secondly, we tested how light intensities affect symbiotic nitrogen competition and the resulting eco-evolutionary dynamics in photosymbioses.

We show that symbiotic nutrient cycling in the cnidarian-algal symbiosis is regulated by resource competition. Increases in labile carbon availability enhance host ammonium assimilation thereby reducing nitrogen availability for its algal symbionts. Increases in algal photosynthesis and associated labile carbon availability thus stimulate nitrogen competition in the symbiosis. Consequently, algal symbiont densities are lowest under conditions optimal for algal photosynthesis and increase toward the environmental tolerance limits of the symbiosis.

## Results and discussion

### Labile carbon availability limits ammonium assimilation in the cnidarian host metabolism

In the cnidarian-algal symbiosis, the host and its algal symbionts have the cellular machinery to use carbon backbones from the tricarboxylic acid (TCA) cycle for amino acid synthesis by assimilating ammonium ($NH_4^+$) via the glutamate metabolism (Fig. 1A)[31,46,47]. Carbon translocation by algal symbionts could thus enhance nitrogen assimilation by their host[24,39,43]. However, the metabolic controls of host ammonium

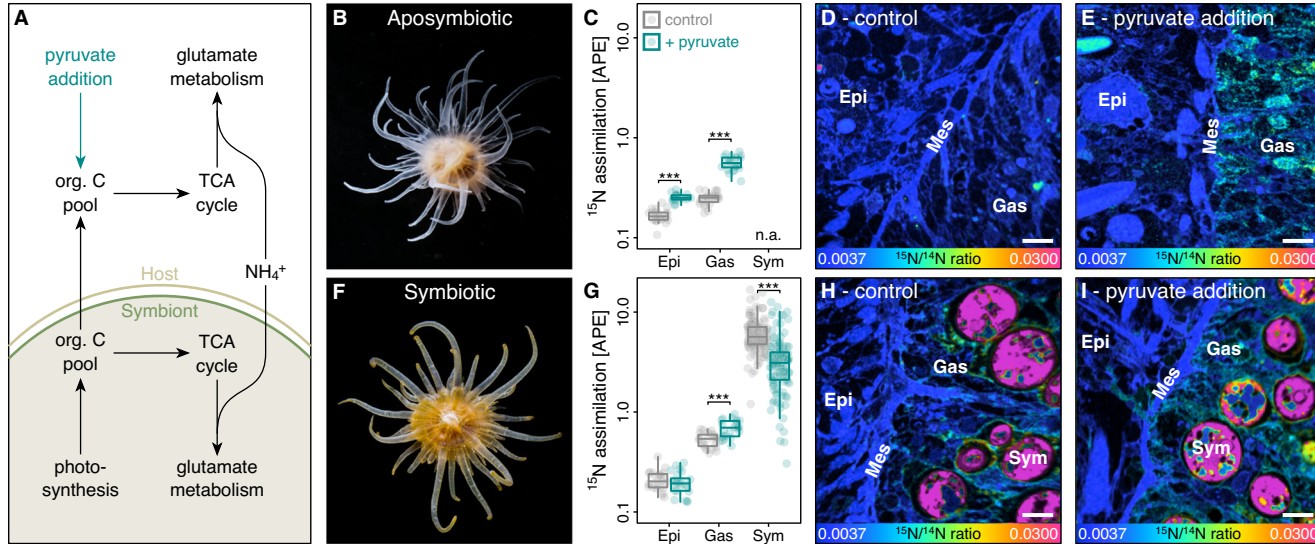

**Fig. 1 | Effects of labile carbon availability on ammonium ($NH_4^+$) assimilation in Aiptasia. A** Amino acid synthesis by the host and its algal symbionts links carbon and nitrogen cycling in the symbiosis. **B** Aposymbiotic Aiptasia were used to quantify (**C**) the effect of 10 mM pyruvate addition on light $^{15}NH_4^+$ assimilation in host tissue layers using (**D, E**) NanoSIMS imaging. Likewise, (**F**) symbiotic Aiptasia were used to quantify (**G**) the effect of 10 mM pyruvate addition on light $^{15}NH_4^+$ assimilation in the host tissue layers as well as algal symbiont cells using (**H, I**) NanoSIMS imaging. Scale bars are 5 μm. Boxplots indicate median, upper and lower quartiles, whiskers show 1.5 × interquartile range. inter Asterisks indicate significant effects of pyruvate addition on $^{15}N$ assimilation in the specific tissue or cell (***$p < 0.001$). 24 host epidermis, 24 host gastrodermis, and 40 algal symbiont regions of interest from three Aiptasia were analyzed per treatment condition for aposymbiotic and symbiotic Aiptasia, respectively. APE atom % excess relative to unlabeled controls, Epi host epithelium, Gas host gastrodermis, Mes host mesoglea, Sym algal symbionts. Source data are provided as a Source Data file.

assimilation and its consequences for algal symbiont nitrogen uptake remain largely speculative[28,36,37,42].

Here, we investigated the effects of 10 mM pyruvate addition on ammonium assimilation in aposymbiotic (algal symbiont-free; Fig. 1B) and symbiotic (algal symbiont-bearing; Fig. 1F) Aiptasia. Previous studies suggest that algal symbionts may lack the cellular machinery to utilize pyruvate[48,49]. Indeed, bulk isotope analysis confirmed that host tissues of both aposymbiotic and symbiotic Aiptasia efficiently assimilated [2,3-$^{13}$C]-pyruvate, while $^{13}$C enrichments of algal symbionts were two orders of magnitude lower (Fig. S1A). Pyruvate addition hence increased labile carbon availability for the host but not the algal symbiont, allowing us to study how the availability of labile carbon in the host metabolism affects ammonium assimilation by symbiotic partners. Consistent with previous observations[50], incubations with 10 μM ammonium revealed that aposymbiotic Aiptasia showed net release of ammonium during 6 h incubations. In contrast, symbiotic Aiptasia showed net uptake of ammonium from the seawater (Tukey HSD, $p < 0.001$). Pyruvate addition, however, increased ammonium uptake by both aposymbiotic and symbiotic Aiptasia (ANOVA, $F = 181.8$, $p < 0.001$), resulting in a depletion of ammonium in the seawater to the limit of detection within 3 h of incubation (Fig. S1B–D). The retention and uptake of ammonium by the Aiptasia holobiont thus appear to be limited by labile carbon availability. Algal photosynthesis or environmental carbon sources (here pyruvate) thus enhance metabolic nitrogen demand in the holobiont, likely resulting in nitrogen limitation for symbiotic partners under oligotrophic conditions.

To disentangle the contribution of symbiotic partners to these changes in holobiont ammonium uptake, we quantified the assimilation of ammonium-$^{15}$N in the host tissue layers and algal symbiont cells following 6 h incubations. NanoSIMS imaging revealed that aposymbiotic Aiptasia showed significantly less ammonium assimilation in their tissues than their symbiotic counterparts (ANOVA, $F = 94.0$, $p < 0.001$; Fig. 1B–I). However, pyruvate addition caused a pronounced increase in aposymbiotic ammonium assimilation (Tukey HSD, $p < 0.001$ for epidermis and gastrodermis, respectively) to $^{15}$N enrichment levels resembling those of symbiotic Aiptasia. Likewise, pyruvate addition also enhanced ammonium assimilation in symbiotic Aiptasia. However, this effect was restricted to the gastrodermal tissue, i.e., the tissue hosting the algal symbionts (Tukey HSD, $p = 0.979$ for the epidermis and $p < 0.001$ for the gastrodermis), potentially reflecting the elevated expression and localization of bi-directional ammonium transporters reported for gastrodermal cells in symbiotic Aiptasia[43].

Importantly, algal symbiont ammonium assimilation showed a reversed pattern with significantly reduced $^{15}$N-enrichment following pyruvate addition (Tukey HSD, $p < 0.001$). These contrasting effects of pyruvate addition on the host and algal symbiont ammonium assimilation likely reflect their contrasting carbon utilization. Our results show that the uptake (and recycling) of ammonium by the host is limited by their access to labile carbon; here pyruvate. While this limitation appears to be most pronounced in aposymbiotic animals, even symbiotic hosts were able to utilize the increased labile carbon availability to stimulate their amino acid synthesis[42]. Consequently, under our experimental conditions (i.e., after 1 week without heterotrophic nutrient sources) algal symbiont photosynthate translocation appears insufficient to fully saturate the metabolic carbon requirements of their host in a stable symbiosis in Aiptasia. The observed reduction in ammonium assimilation by algal symbionts during pyruvate addition is thus likely a direct consequence of increased host anabolic activity. These results suggest that symbiotic partners compete for ammonium and that the outcome of this competition is directly linked to carbon availability.

These findings imply that a positive feedback loop regulates nutrient cycling in the symbiosis: translocation of algal photosynthates

stimulates anabolic ammonium assimilation by their host. Reduced nitrogen availability for algal symbionts limits their growth and further enhances photosynthate translocation to the host. Environmental conditions affecting algal photosynthetic performance and/or organic nutrient pollution could thus directly alter nitrogen availability for algal symbionts because of changes in carbon availability in the host metabolism.

## Algal photosynthesis alters symbiotic nitrogen availability

To test this hypothesis, we performed a long-term experiment on the effects of light availability on symbiotic Aiptasia. Over 6 months, Aiptasia were gradually acclimated to seven different light levels spanning an exponential gradient from near dark (photosynthetically active radiation (PAR < 6.25 μE s$^{-1}$ m$^{-2}$) to high light intensity (PAR = 400 μE s$^{-1}$ m$^{-2}$), and animals were left at their final treatment levels for 1 month without supplemental feeding. The lowest and highest light treatment levels exceeded the tolerance limits of a stable symbiosis. At near-dark conditions, Aiptasia exhibited a near-complete loss of algal symbionts (bleaching) while the host remained viable. At the highest light intensity, Aiptasia still hosted algal symbionts but suffered from severe host mortality (63%). These symbiotic tolerance limits, i.e., PAR from 12.5 to 200.0 μE s$^{-1}$ m$^{-2}$, likely reflect the trade-offs between increasing light limitation toward the lower end of the light gradient and increasing photooxidative stress toward the higher end of the light gradient, respectively[11,21,51,52].

Indeed, the role of photodamage was reflected in the maximum photosynthetic efficiency of algal symbionts declining by more than 20% at the upper tolerance limit (PAR 200.0 μE s$^{-1}$ m$^{-2}$) compared to the lower tolerance limit of the symbiosis (PAR 12.5 μE s$^{-1}$ m$^{-2}$; LM, $F = 62.9$, $p < 0.001$; Fig. 2A). This trade-off between light limitation and excess light stress was evident in the gross photosynthetic activity of individual algal cells showing an optimum response with the highest activities at intermediate light levels, i.e., PAR 50–100 μE s$^{-1}$ m$^{-2}$ (LM, $F = 3.5$, $p = 0.048$; Fig. 2B). Bulk stable isotope measurements of Aiptasia incubated with $^{13}$C-bicarbonate confirmed this pattern as algal cells contributed the most photosynthetically fixed carbon to the holobiont at intermediate light levels (LM, $F = 11.9$, $p < 0.001$; Fig. 2C). In line with this, atomic carbon to nitrogen ratios (C:N ratios) of holobionts followed a similar pattern along the light gradient (LM, $F = 5.1$, $p = 0.016$; Fig. 2D). Given that environmental nitrogen availability, i.e., dissolved seawater nutrients and heterotrophic feeding, was identical for all light treatments, these changes in the C:N ratio likely reflect the direct changes in carbon availability in the holobiont as a function of gross photosynthetic activity. Consequently, our results suggest that labile carbon availability in the symbiosis is highest under intermediate light levels and decreases toward the tolerance limits of the symbiosis at lower and higher light intensities.

The link between carbon availability and symbiotic nitrogen cycling identified above implies that algal photosynthesis should alter nitrogen assimilation in the symbiosis. Indeed, the optimum in photosynthetic carbon fixation and C:N ratios were consistent with a similar pattern of bulk stable isotope measurements of $^{15}$N-ammonium labeled Aiptasia showing the highest ammonium assimilation in the holobiont at intermediate light levels (LM, $F = 4.6$, $p = 0.022$; Fig. S2A). To further disentangle the respective contribution of symbiotic partners to holobiont ammonium assimilation, we used NanoSIMS imaging. This revealed that the optimum in holobiont ammonium assimilation was the composite result of the asymmetric responses of the host and algal symbionts, respectively. On average, algal symbionts showed an approximately ten-fold higher $^{15}$N enrichment than the surrounding host tissue (Fig. 2E–G). However, algal ammonium assimilation peaked at low light availability, i.e., PAR 25–50 μE s$^{-1}$ m$^{-2}$, while host ammonium assimilation in the gastrodermis and epidermis peaked at intermediate light levels, 50–100 μE s$^{-1}$ m$^{-2}$ (Figs. 2E–G

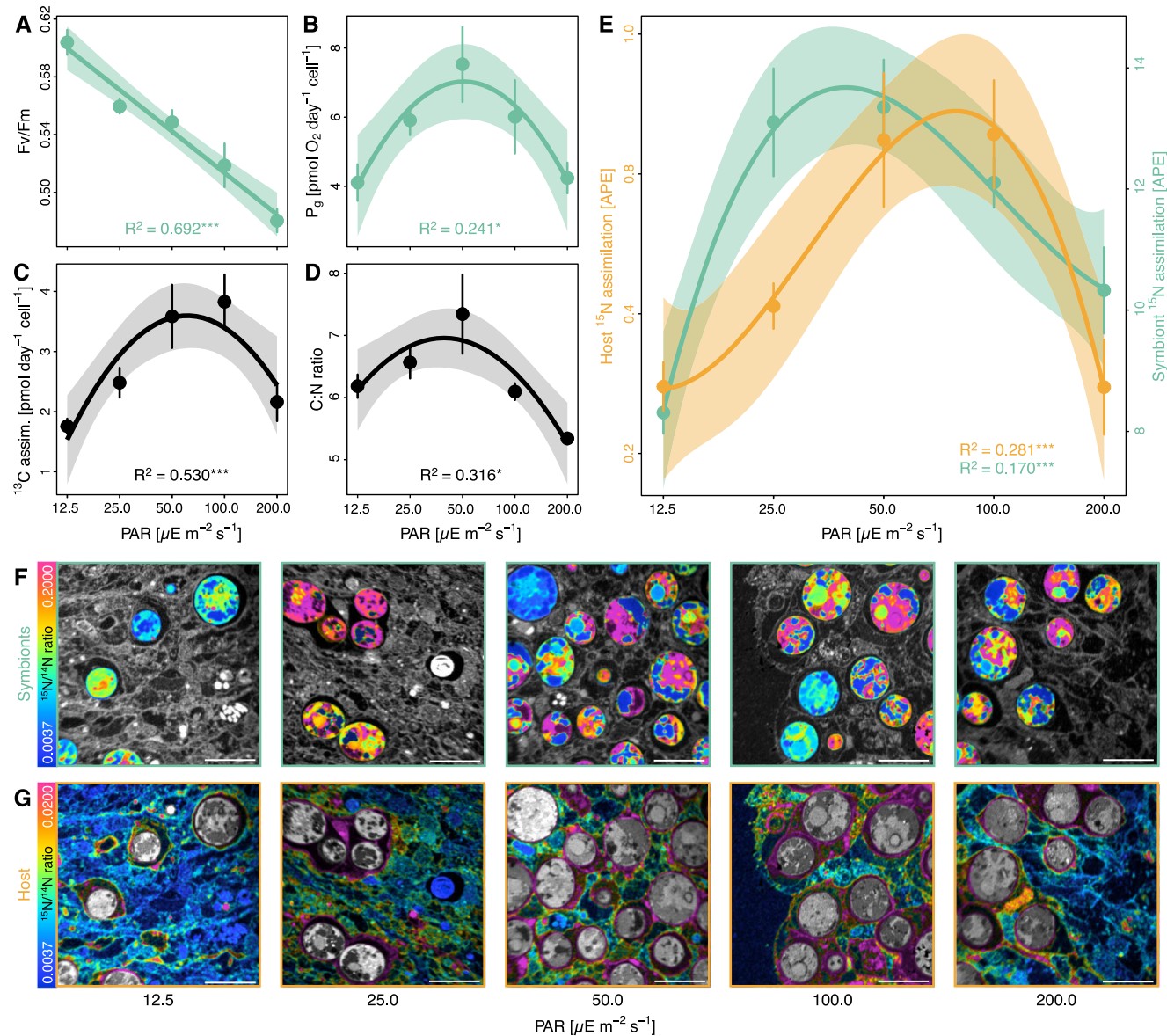

**Fig. 2 | Effects of light availability on symbiotic carbon and nitrogen cycling in Aiptasia.** **A** Maximum quantum yield and (**B**) Gross photosynthesis ($P_g$) of algal symbionts. **C** Bulk $H^{13}CO_3^-$ assimilation of Aiptasia holobionts (i.e., host + algae) normalized by algal symbiont content. **D** Atomic C:N ratios of Aiptasia holobionts. **E** NanoSIMS analysis of $^{15}NH_4^+$ assimilation in the host gastrodermis and algal symbionts. **F** Representative NanoSIMS hue saturation images of $^{15}N/^{14}N$ ratios of algal symbionts with surrounding host tissue shaded in gray. **G** Representative NanoSIMS hue saturation images of $^{15}N/^{14}N$ ratios of host gastrodermal tissue with algal symbionts shaded in gray. Note different scales for (**F**) and (**G**). Scale bars are 10 μm. Points and error bars indicate mean ± SE; lines represent best-fitting models with corresponding confidence intervals. $R^2$ values show the proportion of variance explained by the model, and asterisks indicate a significant effect of light availability on the response parameter (*$p < 0.050$; **$p < 0.010$; ***$p < 0.001$). Per light condition, 6 (**A**) or 5 (**B**–**D**) Aiptasia were analyzed. For NanoSIMS analyses, 16 host and 32 algal symbiont regions of interest were analyzed from two Aiptasia per light condition. APE atom % excess relative to unlabeled controls, PAR photosynthetically active radiation. Source data are provided as a Source Data file.

and S2B; algal symbionts: LM, $F = 10.6$, $p < 0.001$; host gastrodermis: LM, $F = 9.9$, $p < 0.001$; host epidermis: LM, $F = 4.6$, $p = 0.008$). While $^{15}N$ enrichments only reflect the uptake of environmental nitrogen in the symbiosis, the recycling of ammonium from the host catabolism is likely regulated by the same processes and follows the same patterns. High $^{15}N$ enrichments thus indicate an increased nitrogen demand in the symbiosis resulting in reduced production of catabolic ammonium by the host. In this light, the combined C:N ratio and $^{15}N$ enrichment patterns (Figs. 2D, E and S2) imply that symbiotic competition for nitrogen is highest under intermediate light levels and decreases toward the tolerance limits of the symbiosis. Hence, our data suggest that increases in photosynthesis and associated labile carbon availability effectively enhance nitrogen limitation for algal symbionts.

## Nitrogen availability shapes the symbiotic phenotype in Aiptasia

Given that nitrogen limitation controls algal growth in the cnidarian-algal symbiosis[30,31,36,37], the here-observed light-dependent changes in symbiotic nitrogen availability should directly affect the regulation of the symbiosis. Indeed, light availability affected the phenotype of photosymbiotic Aiptasia. Pigmentation and chlorophyll autofluorescence were at their highest levels in Aiptasia at the low and high end of the light tolerance range of the symbiosis (Fig. 3A). However, chlorophyll $a$ content of individual algal cells remained unaffected by light availability suggesting that long-term photo acclimation was driven by other holobiont responses, e.g., changes in host pigmentation (LM, $F = 0.0$, $p = 0.995$; Fig. S3A). Nonetheless, chlorophyll $a$

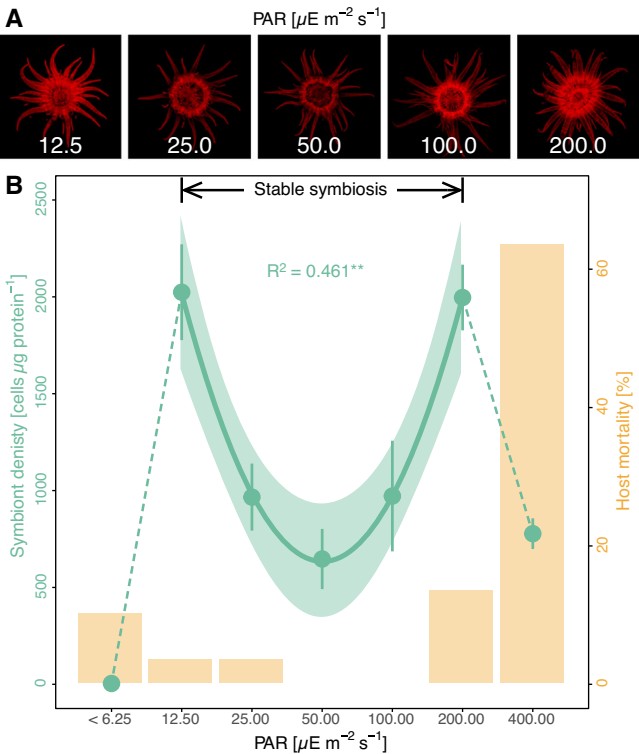

**Fig. 3 | Effects of light availability on the symbioticphenotype of Aiptasia.**
**A** Representative photographs of chlorophyll fluorescence of algal symbionts in Aiptasia depending on light availability. **B** Symbiont density (green) and animal mortality (yellow) depending on photosynthetic active radiation (PAR) levels. Points and error bars indicate mean ± SE; the line represents the best-fitting model with corresponding confidence intervals within the tolerance range of the stable symbiosis. The $R^2$ value shows the proportion of variance explained by the model and asterisks indicate a significant effect of light availability on the response parameter (**$p < 0.010$). Per light condition, gross photosynthesis was quantified for 5 Aiptasia and mortality was assessed across all 30 Aiptasia. Source data are provided as a Source Data file.

content in relation to host protein content increased by more than two-fold toward the symbiosis tolerance limits compared to intermediate light levels (LM, $F = 7.4$, $p = 0.004$; Fig. S3B). This response was driven by changes in algal symbiont densities, which were lowest at $50 \mu E s^{-1} m^{-2}$ and increased by more than three-fold toward the upper and lower light tolerance limit of the symbiosis (LM, $F = 8.9$, $p = 0.002$, Fig. 3B). Consequently, symbiont density was lowest when their gross photosynthetic performance was highest. Indeed, algal symbiont densities showed a pronounced negative correlation with gross photosynthetic activity along the experimental light gradient (Pearson's $r = -0.578$, $p = 0.002$) and holobiont C:N ratios (Pearson's $r = -0.489$, $p = 0.001$) suggesting that the algal symbiont population is regulated by nitrogen availability (as a consequence of resource competition). Under optimal environmental conditions, high photosynthetic carbon availability ensures efficient recycling of catabolic nitrogen in the host metabolism and limits nitrogen availability for algal symbionts. Under suboptimal environmental conditions, reduced carbon availability increases the relative availability of nitrogen in the holobiont, thereby supporting higher symbiont densities, consistent with the feedback loop described above.

**Nutrient cycling controls the eco-evolutionary dynamics of the cnidarian-algal symbiosis**

Our results provide direct experimental support for the role of metabolic interactions in the passive regulation of the cnidarian-algal symbiosis. Specifically, these findings lend support to the bioenergetic

models of symbiosis regulation by Cunning et al.[25] and Cui et al.[39] by highlighting the role of carbon and nitrogen cycling in the eco-evolutionary dynamics of photosymbiotic cnidarians. We found that nitrogen competition between the host and its algal symbionts effectively limits nitrogen availability for algal symbionts under optimal environmental conditions. Toward the environmental tolerance limits of the symbiosis, however, this competition is gradually replaced by nitrogen competition between algal symbionts as their population density increases[38]. This dynamic interplay between inter- and intraspecific nitrogen competition extends the symbiosis' tolerance range by maintaining nitrogen-limited conditions across a wide range of environmental conditions. Indeed, previous reports found that corals with a broad depth distribution harbored the highest symbiont densities at their upper and lower distribution limits, with the lowest symbiont densities occurring toward the center of their distribution range[15,53]. Our results thus suggest that the passive metabolic regulation of symbiotic interactions through resource competition facilitates the ecological success and stability of the cnidarian-algal symbiosis across a wide range of environmental conditions.

Finally, the processes described here improve our understanding of the maintenance of the symbiosis in a changing environment. These mechanisms are unlikely limited to the light-dependent regulation of the symbiosis but likely control other acclimation and stress responses as well. Specifically, our findings may contribute to deciphering the processes leading to the breakdown of cnidarian-algal symbiosis during heat stress[54]. During mass bleaching events, severe and prolonged heat stress causes a dramatic loss of algal symbionts in photosymbiotic Cnidaria, thereby promoting host starvation and reef degradation[55,56]. Yet, algal symbiont densities have been shown to increase prior to bleaching during early and moderate warming, and corals with higher symbiont densities show increased susceptibility to bleaching[34,57–59]. Here, we show that algal symbiont densities may increase toward the symbiosis tolerance limits following reduced carbon availability for the host. Hence, an initial proliferation of algal symbionts during early heat stress is unlikely to be a beneficial response. Instead, increasing symbiont densities in the early phases of environmental stress should be considered a sign of destabilization of symbiotic nutrient cycling, which might ultimately contribute to the breakdown of the symbiosis during heat stress[34,40,60].

Taken together, we conclude that the cnidarian-algal symbiosis is passively controlled by coupling carbon and nitrogen cycling in the symbiosis. Resource competition stabilizes the symbiosis under a wide range of environmental conditions. At the same time, the resulting negative relationship between host and symbiont performance (zero sum) may promote the evolution of parasitic behavior and destabilize the symbiosis in times of rapid environmental change.

## Material and methods

### Animal culture maintenance

All experiments were performed with Aiptasia clonal lineage CC7 (*sensu Exaiptasia diaphana*)[61]. Prior to the experiments, the animals were reared in illuminated growth chambers (Algaetron 230, Photo System Instruments, Czech Republic) at a constant temperature of 20 °C in 2 L clear food containers (Rotho, Switzerland) filled with artificial seawater (35 PSU, Pro-Reef, Tropic Marin, Switzerland). Each week, Aiptasia anemones were fed with freshly hatched *Artemia* nauplii (Sanders GSLA, USA), thoroughly cleaned, and the seawater was exchanged. Photosymbiotic animals harboring their native algal symbiont community dominated by *Symbiodinium linucheae* (subclade A4, strain SSA01) were reared in a 12 h:12 h light-dark cycle with photosynthetic active radiation (PAR) of $50 \mu E m^{-2} s^{-1}$. Further, aposymbiotic animal cultures deprived of algal symbionts were generated following established cold shock bleaching protocols[62]. Briefly, photosymbiotic animals were cold-shocked at 4 °C for 4 h followed by 8 weeks of culturing in artificial seawater containing 50 μM DCMU at constant

irradiance of $50 \, \mu E \, m^{-2} \, s^{-1}$. The absence of algal symbionts was confirmed using fluorescence microscopy and aposymbiotic cultures were maintained at continuous darkness thereafter.

## Experiment 1: effect of pyruvate on nitrogen cycling

To study the effect of labile carbon availability on symbiotic nitrogen cycling, we assessed how pyruvate affects ammonium ($NH_4^+$) assimilation in aposymbiotic and symbiotic Aiptasia. Following 1 week of starvation, 26 aposymbiotic and 26 photosymbiotic animals were transferred into individual 50 ml glass vials for isotope labeling. For the incubations, a minimal artificial seawater medium (35 PSU, pH 8.1, 355.6 mM NaCl, 46.2 mM $MgCl_2$, 10.8 mM $Na_2SO_4$, 9.0 mM $CaCl_2$, 7.9 mM, KCl, 2.5 mM $NaHCO_3$) containing $10 \, \mu M$ $^{15}NH_4Cl$ (≥98 atom % $^{15}N$) was used following Harrison et al.[63]. While half of the animals (13 aposymbiotic and 13 photosymbiotic) were incubated in minimal artificial seawater medium, the other half was incubated in minimal artificial seawater medium spiked with 10 mM [2,3-$^{13}C$]-pyruvate (99 atom % $^{13}C$). Incubations were performed using the culturing conditions outlined above, with aposymbiotic animals being kept in the dark and photosymbiotic animals being kept in constant light. After 3 h, the 5 incubations were terminated for each combination of treatment and symbiotic state and the incubation water was collected to quantify ammonium concentrations (see below). After 6 h, the remaining incubations were terminated. For each combination of treatment and symbiotic state, the water was collected from 5 incubations for ammonium measurements, and Aiptasia were sampled for either bulk isotope measurements (5 animals) of $^{13}C$ assimilation or NanoSIMS analysis (3 animals) of $^{15}N$ assimilation (see below).

## Experiment 2: effect of light on nitrogen cycling

To study the effect of light on symbiotic nitrogen cycling, photosymbiotic animals were kept in an exponential series of light levels in a long-term experiment. All seven light treatments followed a 12 h:12 h light-dark cycle and used the same culturing conditions outlined above. Over the course of 6 months, 30 animals per treatment were gradually acclimated in monthly steps to the following PAR levels: <6.25, 12.50, 25.00, 50.00, 100.00, 200.00, $400.00 \, \mu E \, m^{-2} \, s^{-1}$. Following acclimation, animals were starved for 1 month to minimize confounding feeding responses on symbiotic nitrogen cycling. After this, the rate of survival of animals was recorded (any asexual offspring were removed during the weekly cleaning routines throughout the experiment). Aiptasia showed severe bleaching at low light levels while hosts suffered from high mortality at high light levels, respectively. Hence, only host mortality and symbiont densities were quantified for all light treatments. All other parameters were only recorded within the tolerance range of the stable symbiosis, i.e., PAR 12.5–$200.0 \, \mu E \, m^{-2} \, s^{-1}$. To quantify bicarbonate and ammonium assimilation in the symbiosis, five animals per treatment were incubated in 50 ml minimal artificial seawater medium containing 2.5 mM $NaH^{13}CO_3$ (≥98 atom % $^{13}C$) and $10 \, \mu M$ $^{15}NH_4Cl$ (≥98 atom % $^{15}N$) for 24 h at their respective treatment conditions. Following the incubation, tentacles from two animals were dissected and fixed for NanoSIMS analysis, and all animals were immediately collected for bulk elemental and isotope analysis (see below). Further, the dark-adapted photosynthetic efficiency and oxygen fluxes were recorded for six and five animals per treatment, respectively (see below).

## Ammonium uptake

Collected seawater samples were immediately filtered (PES, $0.22 \, \mu m$), transferred into sterile 15 ml centrifuge tubes, and stored at −20 °C for subsequent analysis. Within 1 week of sampling, samples were defrosted, and ammonium concentrations were immediately analyzed using a Smartchem450 wet chemistry analyzer (AMS Alliance, Italy). Changes in ammonium concentrations were corrected for volume and duration of incubations, and fluxes were normalized to the dry weight of animals (freeze-dried after carefully removing any excess water using filter paper).

## Fixation, embedding, and NanoSIMS imaging

Collected animals (experiment 1) and individual tentacles (experiment 2) were rinsed in artificial seawater without isotope tracers and immediately transferred into a fixative solution (2.5% glutaraldehyde and 1% paraformaldehyde in 0.1 M Sorensen's phosphate buffer) and incubated for 1 h at room temperature followed by 24 h at 4 °C. Samples were transferred into a storage solution (1% paraformaldehyde in 0.1 M Sorensen's phosphate buffer) and processed for dehydration and resin infiltration within 4 days of collection. Samples were incubated for 1 h in 1% $OsO_4$, rinsed, and dehydrated in a series of increasing ethanol concentrations (30% for 10 min, 50% for 10 min, 2 × 70% for 10 min, 3 × 90% for 10 min, and 3 × 100% for 10 min). For resin infiltration, samples were transferred to 100% acetone and gradually infiltrated with Spurr resin (Electron Microscopy Sciences, USA) at increasing concentrations (25% for 30 min, 50% for 30 min, 75% for 1 h, and 100% overnight), and resins were cured for 48 h at 65 °C. Embedded samples were cut into 200 nm thin sections with an Ultracut E ultra-microtome (Leica, Germany), transferred onto glow-discharged silicon wafers, and coated with a 12 nm gold layer.

The surface isotopic composition of sample sections on silicon wafers was analyzed using a NanoSIMS 50 L (Cameca, France)[64]. Following pre-sputtering for 5 min with a primary beam of ca. 6 pA to remove the metal coating, samples were bombarded with a 16 keV primary ion beam of ca. 2 pA $Cs^+$ focused to a spot size of about 150 nm on the sample surface. Secondary molecular cyanide ions $^{12}C^{14}N^-$ and $^{12}C^{15}N^-$ were simultaneously collected in electron multipliers at a mass resolution of about 9000 (Cameca definition), sufficient to resolve the $^{12}C^{15}N^-$ ions from potentially problematic interferences. Eight to nine sample areas were analyzed for each of the samples by rastering the primary beam across a $40 \times 40 \, \mu m$ sample surface with a $256 \times 256$ pixels resolution and a pixel dwell time of 5 ms for five consecutive image layers. The resulting isotope images were processed using the ImageJ plug-in OpenMIMS (https://github.com/BWHCNI/OpenMIMS/wiki). Mass images were drift and dead-time corrected, the individual planes were summed, and the $^{12}C^{15}N^-/^{12}C^{14}N^-$ ratio images were expressed as a hue-saturation-intensity image, where the color scale represents the $^{15}N/^{14}N$ isotope ratio. $^{15}N$ assimilation was quantified by drawing regions of interest (ROIs) of host epidermis, host gastrodermis, and algal symbionts based on $^{12}C^{14}N^-$ images, respectively (Fig. S4). As individual host cells were not clearly distinguishable in the NanoSIMS images, all epidermal and gastrodermal tissue areas within one image (excluding algal symbionts and symbiosomal contents) were recorded as one ROI per tissue layer, respectively (Fig. S4). For experiment 1, this yielded 8 epidermal and gastrodermal ROIs per Aiptasia respectively. For experiment 2, this yielded 4 epidermal and 8 gastrodermal ROIs per Aiptasia. Algal symbionts ROIs were drawn based on individual algal cells and only the largest ROIs (40 ROIs per Aiptasia in experiment 1 and 16 ROIs per Aiptasia in experiment 2) were included in the analysis to minimize potential measurement variability due to lower signal-to-noise ratio of smaller ROIs. Algal symbiont ROI size thus had no significant effect on $^{15}N$ enrichments (LM, $F = 1.5$, $p = 0.224$ for experiment 1, $F = 0.1$, $p = 0.720$ for experiment 2; Fig. S5). For each ROI, $^{15}N$ enrichment was expressed as atom % express (APE) relative to unlabeled controls. Notably, these enrichment values are likely an underestimation of the actual enrichment levels for these organisms as fixation, dehydration, and resin embedding during sample preparation extract and dilute soluble compounds from the sample matrix. However, any methodological bias arising from this is consistent across samples, NanoSIMS images hence allow for a robust assessment of relative enrichment values. In light of the clonal nature of Aiptasia and the identical environmental conditions of animals within the same treatment, individual ROIs were considered as

independent measurements at the microscale level regardless of the animal replicate for the purpose of the analysis.

## Bulk elemental and isotope analysis

Anemones were collected following stable isotope labeling, rinsed in artificial seawater without isotope tracers, and immediately homogenized in 1 ml MilliQ water using a Polytron PT1200E immersion dispenser (Kinematica, Switzerland). For experiment 1, host and algal symbiont fractions were separated by centrifugation (1000 × $g$ for 5 min, sufficient to pellet >95% of algal symbionts from the sample), the algal pellet was rinsed twice by resuspension and centrifugation, and both fractions were snap-frozen in liquid nitrogen and kept at −80 °C until further analysis. For experiment 2, homogenized samples were immediately snap-frozen and kept at −80 °C without separating host and algal symbiont fractions. All samples were freeze-dried, and the dry mass was recorded. The atomic carbon (C) to nitrogen (N) ratios were quantified using a Carlo Erba 1108 elemental analyzer (Fisons Instruments, Italy). This was coupled via a Conflo III interface to a Delta V Plus isotope ratio mass spectrometer (Thermo Fisher Scientific, Germany) to determine the carbon (experiments 1 and 2) and nitrogen (experiment 2) stable isotope composition. $^{13}C/^{12}C$ and $^{15}N/^{14}N$ ratios were calibrated with six in-house urea standards with defined isotope ratios as described in Spangenberg and Zufferey[65] and normalized against the international Vienna Pee Dee Belemnite limestone (VPDB) and Air-N$_2$ scale, respectively. To assess the isotope enrichment of samples, the isotope ratios were converted to atom % and converted to APE by subtracting isotopic values measured in unlabeled control animals. For experiment 2, absolute isotope tracer assimilations of Aiptasia holobionts were normalized to their algal symbiont content to account for potential differences in symbiont densities between samples.

## Photosynthetic efficiency and gross photosynthesis

To assess the effect of light availability on the photosynthetic efficiency of algal symbionts, pulse amplitude modulated (PAM) fluorometry was used. Anemones were dark-acclimated for 1 h before the maximum quantum yield (Fv/Fm) was recorded using the blue light version of the Mini-PAM-II (Walz, Germany). For each specimen, the initial fluorescence (Fo) was recorded, followed by a saturating pulse, and maximum fluorescence (Fm) was measured directly afterward. The maximum quantum yield was calculated as the ratio between the variable fluorescence (Fv = Fm − Fo) and the maximum fluorescence.

To quantify gross photosynthetic rates of the anemones, respiration, and net photosynthesis were measured in separate consecutive incubations. For this, anemones were transferred into 12.3 ml borosilicate vials filled with artificial seawater and equipped with an OXSP5 oxygen sensor spot (Pyroscience, Germany). Following the attachment of the animals at the vial bottom, a 6 mm magnetic stirrer was added, and each vial was sealed bubble-free, inverted, and transferred to a water bath. Vials were continuously stirred at 240 rpm using a magnetic stirring plate, and oxygen concentrations were continuously recorded with an FSO2-4 oxygen meter (PyroScience) connected via a SPFIB-BARE optical fiber (PyroScience). All incubations were performed at 20 °C for ~2 h each. Respiratory oxygen consumption was quantified during dark incubations. Subsequently, net photosynthetic activity was recorded during light incubations with light levels according to the respective treatment conditions of the animals. Oxygen fluxes were corrected for artificial seawater control incubations and normalized to the incubation duration and the animals' algal symbiont content (see below). Gross photosynthesis was calculated as the sum of the net photosynthesis rate and the absolute value of the respiration rate of each animal. Notably, this method does not account for potential increases in respiration rates during the light[66]. The here presented gross photosynthesis rates thus likely represent an underestimation of the actual rates. However, this methodological bias is consistent across treatments and does not impair the conclusions presented here.

## Algal symbiont densities, chlorophyll a content, and protein content

Following oxygen flux incubations, anemones were immediately homogenized in 0.5 ml 2x PBS using a Polytron PT1200E immersion dispenser (Kinematica), and host and symbiont fractions were separated by centrifugation (1000 × $g$ for 5 min, sufficient to pellet >95% of algal symbionts). The host supernatant was stored at −20 °C until further analysis. The algal symbiont pellet was resuspended in 1 ml 2x PBS and divided into two equal aliquots. The first aliquot was used to quantify the concentration of algal symbiont cells in six replicate measurements using a Countess cell counter Countess II FL Automated Cell Counter (Thermo Fisher Scientific, USA) with a Cy5 light cube. Algal symbiont cells could be clearly identified by the cell counter based on their size, roundness, and chlorophyll fluorescence. The second aliquot was used for chlorophyll $a$ content quantification. For this, cells were pelleted (5000 × $g$ for 5 min), resuspended in 90% ethanol, and incubated at 4 °C for 24 h in the dark under constant agitation. After this, cell debris was pelleted (5000 × $g$ for 5 min), and two 200 µl supernatant per sample were transferred into a transparent flat-bottom 96-well plate. Sample absorbances were immediately recorded at 630, 664, and 750 nm using a CLARIOstar Plus microplate reader (BMG Labtech, Germany), and chlorophyll content was calculated according to Jeffrey and Humphrey[67]:

$$\text{Chlorophyll } a[\mu g\, mL-1] = 11.43 \times (OD664 - OD750) \\ - 0.644 \times OD630(-OD750)$$

Symbiont densities and chlorophyll $a$ content were normalized against host tissue protein content as a proxy of host biomass. Protein content in the defrosted host supernatant was quantified according to Bradford[68]. Three technical 5 µl replicates were analyzed for each sample using the Pierce Coomassie (Bradford) Protein Assay Kit (Thermo Scientific, US) according to the manufacturer's instruction, and the absorbance of samples, as well as BSA calibration standards, was recorded at 595 nm using a CLARIOstar Plus microplate reader (BMG Labtech).

## Statistical analysis

For the first experiment, NanoSIMS data from the host and symbiont compartments were square root transformed to meet model assumptions analyzed separately. Bulk isotope, ammonium flux, and NanoSIMS measurements were analyzed with an analysis of variance (ANOVA) based on symbiotic state (aposymbiotic vs. photosymbiotic) and treatment (control vs. pyruvate addition) followed by Tukey's honest significance test. Changes in Ammonium concentrations over time were analyzed using an analysis of variance (ANOVA) based on symbiotic state (aposymbiotic vs. photosymbiotic) and treatment (control vs. pyruvate addition) and time point (3 h vs. 6 h) followed by Tukey's honest significance test.

For the second experiment, only measurements within the defined tolerance range of the stable symbiosis, i.e., PAR 12.5–200.0 µE m$^{-2}$ s$^{-1}$, were used for the statistical analysis. All data were analyzed using linear regression models (LM) using light treatment levels as an explanatory variable. Where necessary, a second-degree (symbiont density, chlorophyll $a$ content normalized by host protein, gross photosynthesis, C:N ratio, bulk $^{13}C$ and $^{15}N$ assimilation) or third-degree (NanoSIMS measurements for host tissues and symbiont cells) polynomial transformation was applied to the data to meet model assumptions. Further, Pearson correlations were used to test the linear relationship between gross photosynthesis rates and symbiont densities.

For both experiments, the effects of algal symbiont ROI size on algal $^{15}N$ enrichments were analyzed using LM using the number of pixels per algal ROI as the explanatory variable.

**Reporting summary**

Further information on research design is available in the Nature Portfolio Reporting Summary linked to this article.

## Data availability

All associated data including NanoSIMS and fluorescence images are available at https://doi.org/10.5281/zenodo.7404864. Source data are provided with this paper.

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

## Acknowledgements

We are grateful to Kristin Bär-Hage, Myriam Schmid, and Petra Merkel for their support with culture maintenance at the University of Konstanz. Dr. Michael Laumann and Dr. Paavo Bergmann from the Electron Microscopy Centre at the University of Konstanz are thanked for their assistance with sample preparation. Further, we thank Karine Vernez and Sylvain Coudret from the Central Environmental Laboratory at EPFL for their help with ammonium measurements. N.R. and A.M. were supported by the Swiss National Science Foundation, grants 200021_179092 and 205321_212614.

## Author contributions

N.R. conceived and conducted the experiment. N.R., S.E., and J.E.S. performed sample and data analysis. N.R., S.E., J.E.S., C.R.V., and A.M. contributed to writing and revising the manuscript.

## Competing interests

The authors declare no competing interests.
