## [Peer Review File · Nature Communications]

Coupled carbon and nitrogen cycling regulates the cnidarian-algal symbiosisREVIEWER COMMENTS

Reviewer #1 (Remarks to the Author):

Review: Coupled carbon and nitrogen cycling regulates cnidarian-algal symbiosis

Rädecker et al. present a two experiment study focused on a model sea anemone (*Aiptasia*) and its symbiotic algal partners (*Symbiodinium linuchae* str SSA01). Two experiments were performed using stable isotope labeling of $^{15}\text{NH}_4$ and NanoSIMS imaging for measuring the ammonium uptake in both partners. The overarching goal was to improve the understanding of the regulation between the partners as it relates to nitrogen limitation and carbon availability. One experiment assessed the effect of labile carbon by addition of pyruvate, and in the second experiment, the light regime was altered from low to high intensities. The key finding of the first experiment was that the partners are competing for N, while in the second experiment, the competition between the partners persists under optimal conditions, and switches to only among the symbionts at the extremes (e.g. high and low light). The authors argue the switch in competition from inter to intra allows a more stable mutualism across a broader range of conditions.

The experiments were performed well, and analyses of the results are conclusive. One weakness in the experimental design was to not utilize a ^{13}C labelled pyruvate, however, the authors likely have an explanation.

The significance of the work is that it provides an important explanation for stabilizing a model representative of cnidarian-algal symbioses. These populations in the wild play an important role in the coral reef ecosystems, and hence the work contributes new information. It seems the authors have a convincing dataset with multiple measurements that should be worthy input (photosynthesis, biosynthesis, respiration, etc.) for a hypothetical flux model. Moreover, this flux model could be used to test different future climate conditions expected for reefs. Perhaps outside the scope or intention of the manuscript, it could be something to consider. A few suggestions and comments are provided with the intention of improving the presentation. The article could also benefit from the help of a native speaker to edit or review the text.

Abstract

-It seems appropriate to mention a bit more details on which form of nitrogen e.g. e.g. $^{15}\text{NH}_4$

Line 34, add "stable" before the isotope

General comment. The first sentence of the abstract is difficult to comprehend, consider to read aloud and re-write. .. Efficient nutrient cycling underpins the ecological success of cnidarian-algal symbioses which thrive in oligotrophic waters and represent the foundation of coral reef ecosystems.

Introduction.

General comment. The introduction provides the details necessary for the role of nutrient exchange in the symbiotic system, but lacks an equal attention on light intensity. Given the second experiment focuses on light, perhaps the introduction should provide some information on the current state of knowledge for light intensity on the symbiotic system.

L54- phrase "shine a light" could be replaced with "could elucidate the evolutionary..."

General comment. Can the end of the introduction strengthen the rationale and motivation for the general audience; for example, besides the fact that the factors are poorly understood for nutrient regulation in cnidarian-algal symbioses, is there a larger motivation/impact for this ...e.g. its relevance to the environment and/or future expected conditions.

Results & Discussion

L 87 -why have the authors not considered a multi label experiment of ^{13}C -pyruvate and $^{15}\text{NH}_4$? The nanoSIMS 50L can measure both up to 6 masses simultaneously (e.g. ^{13}C -, ^{12}C -, $^{12}\text{C}^{14}\text{N}$ -, $^{12}\text{C}^{15}\text{N}$ -). It seems like a shortcoming in the experimental design, but perhaps there is an explanation?

L90. The use of "and/or" which is it-the algal symbionts have limited access and lack

machinery OR have limited access or lack machinery? This detail seems rather crucial to the experimental design.

L92. What is the label percent of $^{15}\text{NH}_4^+$? This is important for the SIMS measures, and should be described here or in the materials and methods.

L107- Can the authors clarify how it is known that the photosynthetic hosts can utilize the additional carbon for their amino acid synthesis. Is this from previous works or the experiment described in the study?

Fig 1 explain what APE is in the caption, add in the number of ROIs or cells analyzed-this detail should also be added to the materials and methods.

L138-140. Seems like this speculation/interpretation could use a reference? Or a comparison to another study?

General comment: With the data acquired in cell abundances and chl a content, is there any indication that the cells are altering their chl a content relative to their cell size? e.g. under high/low light algal cells can alter the pigment content for energy conservation or dissipation.

Materials and methods

L 277-278. Rephrase, seems like host mortality is not necessary in the first part of the sentence.

General comment. Fixation procedures could lead to a dilution of the enrichment, the authors could consider making a statement in the methods section to explain this. Several papers have reported this for bacteria, perhaps similar response is valid for the host and algal cells.

L309. It is probably better to state that the $^{15}\text{N}/^{14}\text{N}$ ratios were based on the $^{12}\text{C}^{15}\text{N}/^{12}\text{C}^{14}\text{N}$.

L312. I think “ion images” is a better descriptor than “maps.”

L329-330. Have the authors also tried to normalize to time in order to report rate?

Figures

General comment. Authors should consider to move and combine Fig S1 and Fig S2 to replace Fig. 3. These two supplementary figures are quite clear overview of the results from Exp. 2.

Suggestion/Question on analyses. Given that nanoSIMS is an amazing MSI technique with high lateral resolution, have the authors tried to draw and compare regions of high enrichment in the individual Symbiodinium. In Fig S2, a striking observation is the variation in the enrichment within the subpopulations of Symbiodinium in the sections—one can see clusters of cells of high enrichment co-occurring with cells of less enrichment. Furthermore, within a Symbiodinium smaller areas of extreme enrichment e.g., are these representative of storage granules? Is this variation due to variable physiology amongst the Symbiodinium OR related to the depth of the sectioning into the Symbiodinium? This is an important technical caveat in SIMS imaging on populations of cells embedded (or in colonies, etc), as they are not all in the same orientation, and hence one should comment/explain this variation. One could measure the diameter of the Symbiodinium, and plot the enrichment as a function of the cell diameter. This figure also makes me wonder how the ROIs for the host tissue were made? Perhaps the authors can show this as a Suppl. figure as well.

Reviewer #2 (Remarks to the Author):

The cnidarian-phototroph symbiosis in principle relies on the transfer of photosynthetically-derived carbon from the symbiont to the host, while the host provides ideal conditions for the symbiont to thrive. It has been shown previously that increased nitrogen availability can lead to increased symbiont growth and subsequent decreases in the carbon transfer to the host (e.g. O’Neil and Capone, 2008) suggesting a tight coupling and/or host regulation of the symbiotic metabolism under stable environmental conditions. Here, Rådecker et al. present single-cell ammonium uptake by an anemone host (*Aiptasia*) and their photosynthetic symbionts under different carbon availabilities, accomplished by either additions of

pyruvate or changes in light (and resulting changes in photosynthesis by the symbionts). The key finding is that when pyruvate is added to incubations of symbiotic *Aiptasia*, the ammonium uptake of the host increases while the ammonium uptake of the symbiont decreases (Fig. 1G). The authors argue that under these increased external (organic) carbon availabilities (i.e. the pyruvate addition), the host competes for ammonium with its symbionts, leading to the decreased ammonium uptake by the symbionts. This conclusion would indicate that the host could, to some extent, control the availability of nitrogen to the symbionts and the subsequent transfer of carbon to themselves.

I see one major issue with the authors' argumentation. While I do not necessarily disagree that the host may have the capacity to control symbiont growth (via coupling of carbon and nitrogen cycling), the authors do not provide enough evidence of the 'resource competition' between host and symbiont or that nitrogen availability was different for host and symbiont. In more detail, the authors used only one concentration of ammonium in their experiments, i.e. ten micromolar ammonium, and do not show that i) this concentration was limiting during the incubation or ii) that ammonium was used up during the incubation. Generally, competition for a resource, such as nitrogen, should only occur if substrate availability is limiting. However, the authors do not provide evidence that this is the case here. In contrast, the ten micromolar ammonium concentration is substantially more than typical in situ concentrations in the oligotrophic conditions of coral reef ecosystems. In addition, incubations using ¹³C-labeled compounds could have possibly provided the means to constrain the coupled carbon-nitrogen cycling but such data is not provided here. These types of incubations/approaches could have provided additional information on what type of carbon and nitrogen cycling was occurring, and how strongly carbon and nitrogen cycling were coupled. In summary, this manuscript does not provide enough evidence for the authors' conclusions.

Additional comments:

I 37: Please clarify what is meant by 'enhanced symbiotic competition'

I 39: Please clarify what is meant by 'tolerance limits of the symbiosis'. While this is

explained later in the manuscript, I think it is essential that the reader already understands in the abstract what is meant.

I 83/86: I don't think that ammonium is 'fixed'; rather it is taken up and assimilated or incorporated. Using the term 'fixing' or 'fixation' may mislead the reader to think that N₂ fixation plays a role here.

I 107: How do you know whether the pyruvate was used for amino acid synthesis and not for, for example, lipid biosynthesis?

I 176: Doesn't the light mediate photosynthesis and therefore carbon rather than N availability?

Fig. 1: It would be good to have some microscopic context to the nanoSIMS images (and/or outlines of tissues and so on).

Fig. 1: The panels H and I in Fig. 1 do not seem to show substantial differences in the ¹⁵N-enrichment in the symbionts between the control and +pyruvate treatment. They do not seem to reflect the findings shown in panel G.

Fig. 1: In panel C, I would rather show the 'Epi' and 'Gas' data on top of the 'Epi' and 'Gas' data in label G, highlighting that there are no symbionts in the aposymbiotic *Aiptasia*.

References:

O'Neil, Judith M., and Douglas G. Capone. "Nitrogen cycling in coral reef environments." *Nitrogen in the marine environment* (2008): 949-989.

Reviewer #3 (Remarks to the Author):

In this manuscript, Raedecker and colleagues investigate how organic C availability shapes competition for nitrogen and regulates symbiont abundance in the model photosymbiotic system *Aiptasia* (*Exaiptasia diaphana*) and its phototrophic *Symbiodinium* symbionts. The

authors compare ^{15}N -ammonium assimilation under labile C-replete vs. C-deplete conditions, in symbiotic and aposymbiotic *Aiptasia* on single cell level using nanoSIMS. Furthermore, they track symbiont abundance and activity under varying light regimes, leading to differing availabilities of organic C from photosynthates for the host. This manuscript sheds light on the mechanism sustaining (and destabilizing) photosymbiotic systems, and adds to the growing body of literature highlighting nitrogen availability as key factor for this type of symbiosis. While I am not an expert in photosymbiosis, I very much enjoyed reading this interesting and well written manuscript and particularly liked the elegant experiments and combination of methods from single cell to bulk activity measurements. I do have some comments on the manuscript, mainly pertaining to the methodological setup (e.g. the rather high concentration of organic C used in this study), as detailed below.

Main points:

- I would like to hear the authors' opinion on the transferability of the labile C experiments to in situ conditions. The used labile C concentration (10 mM pyruvate) seems rather extreme compared to the natural conditions these animals live in (typical surface seawater DOC concentration is 100-500 μM). I wonder whether the highly elevated pyruvate concentrations used here might distort the host (and symbiont) responses in ^{15}N -assimilation reported in the manuscript. Was the response to lower (more natural) pyruvate additions tested?
- To me it is unclear how many replicates were measured on the nanoSIMS. Please state this in the figure legend (fig 1) and in the main text. This is important to know, as it can otherwise not be assessed how much of the observed differences between +/- pyruvate is due to biological heterogeneity of individual animals, or response to the treatment.
- Likewise, I would suggest to show data from individual *Aiptasia* as separate points in Fig 2, 3, and the supplementary figures, to show variability between individual animals.

Minor comments:

Introduction:

- I suggest to briefly introduce the photosymbiotic system (species) used here, and especially mention the Symbiodiniaceae symbionts by name.

Results & Discussion:

- L100: Do the authors have an explanation for the lack of significant response to pyruvate addition in the gastrodermal tissue of the photosymbiotic animals? Higher general labile C-availability due to the physical proximity to the symbionts and thus lack of response to elevated external C?
- Can the authors make a rough calculation on how much of the added ^{15}N -ammonium was consumed by the holobiont in the 6h (pyruvate experiment) and 24h (light levels experiment)? Was there residual ammonium available?
- Figure 2: In the legend, descriptions of panel A and B are swapped. Please explain the abbreviation APE in the legend (atom percent excess), this is also missing in the other figures depicting nanoSIMS or isotope enrichment data. Please double check the data sheet for the nanoSIMS data of experiment 2 – AP and APE columns contain identical values.

Methods:

- L313-316: I don't understand what is meant by this statement. Please specify how ROIs of the host were drawn. The data sheet suggests that the areas of host ROIs were much larger than the symbiont ROIs – I find this surprising, as I would have intuitively drawn ROIs around individual host cells, which should not be that much larger than the symbionts, to depict the variability in host tissue enrichment.
- Bulk elemental analysis: Please specify how *Aiptasia* individuals were washed before elemental analysis (I assume they were washed, else, ^{15}N ammonium likely sticks to the biomass and biases the measurement)
- L347-352: Is the respiration rate of the host identical in light and dark conditions? This would need to be the case to be able to calculate gross photosynthesis the way it was done. This could be at least roughly assessed by measuring oxygen consumption by aposymbiotic *Aiptasia* in light vs. dark conditions.
- L354-374: Can the authors confirm that no (or very few) symbiont cells are retained in the supernatant after pelleting?

Response to reviewer comments

Coupled carbon and nitrogen cycling regulates the cnidarian-algal symbiosis

Comments by the reviewers are indicated in **black**.

Responses by the authors are indicated in **blue**.

New and revised manuscript sections are indicated in **green**.

Reviewer #1

R1: Räddecker et al. present a two experiment study focused on a model sea anemone (*Aiptasia*) and its symbiotic algal partners (*Symbiodinium linuchae* str SSA01). Two experiments were performed using stable isotope labeling of $^{15}\text{NH}_4$ and NanoSIMS imaging for measuring the ammonium uptake in both partners. The overarching goal was to improve the understanding of the regulation between the partners as it relates to nitrogen limitation and carbon availability. One experiment assessed the effect of labile carbon by addition of pyruvate, and in the second experiment, the light regime was altered from low to high intensities. The key finding of the first experiment was that the partners are competing for N, while in the second experiment, the competition between the partners persists under optimal conditions, and switches to only among the symbionts at the extremes (e.g. high and low light). The authors argue the switch in competition from inter to intra allows a more stable mutualism across a broader range of conditions.

A: Excellent summary! We are delighted to see that we were able to convey the key messages. Thank you for taking the time to review the work and for your excellent suggestions.

R1: The experiments were performed well, and analyses of the results are conclusive. One weakness in the experimental design was to not utilize a ^{13}C labelled pyruvate, however, the authors likely have an explanation.

A: Thank you for raising this valid point. We have now included additional bulk isotope measurements of ^{13}C assimilation for both experiments. These additional analyses confirm that algal symbionts do not assimilate pyruvate, thereby supporting our interpretation of the results (please refer to Figure S1, as well as the revised results/discussion which now includes:

L107: “Previous studies suggest that algal symbionts may lack the cellular machinery to utilize pyruvate (48, 49). Indeed, bulk isotope analysis confirmed that host tissues of aposymbiotic and symbiotic *Aiptasia* efficiently assimilated $[2,3-^{13}\text{C}]$ -pyruvate, while ^{13}C enrichments of algal symbionts were two orders of magnitude lower (**Fig. S1A**). Pyruvate addition hence increased labile carbon availability for the host but not the algal symbiont, allowing us to study how the availability of labile carbon in the host metabolism affects ammonium assimilation by symbiotic partners.”

R1: The significance of the work is that it provides an important explanation for stabilizing a model representative of cnidarian-algal symbioses. These populations in the wild play an important role in the coral reef ecosystems, and hence the work contributes new information. It seems the authors have a convincing dataset with multiple measurements that should be worthy input (photosynthesis, biosynthesis, respiration, etc.) for a hypothetical flux model. Moreover, this flux model could be used to test different future climate conditions expected for reefs. Perhaps outside the scope or intention of the manuscript, it could be something to consider. A few suggestions and comments are provided with the intention of improving the presentation. The article could also benefit from the help of a native speaker to edit or review the text.

A: Good point. We agree that modeling fluxes in the symbioses would be the best way to transfer the findings of this work to other settings. Unfortunately, NanoSIMS analyses are an excellent tool to quantify relative changes in assimilation but do not provide a reliable assessment of absolute fluxes. For the time being, we thus point the reader to the previous excellent work of Cuning et al. 2017 *J Theo Biol* and hope to develop robust flux models for future applications.

The manuscript now states:

L251: "Specifically, these findings lend support to the bioenergetic models of symbiosis regulation by Cunning *et al.* (25) and Cui *et al.* (39)..."

Abstract

R1: -It seems appropriate to mention a bit more details on which form of nitrogen e.g. e.g. ^{15}N

A: The abstract now clearly states that we used ^{13}C and ^{15}N labeling to study the link between ammonium uptake and labile carbon availability.

R1: Line 34, add "stable" before the isotope

A: Done.

R1: General comment. The first sentence of the abstract is difficult to comprehend, consider to read aloud and re-write. .. Efficient nutrient cycling underpins the ecological success of cnidarian-algal symbioses which thrive in oligotrophic waters and represent the foundation of coral reef ecosystems.

A: Agreed. We have removed the reference to coral reef ecosystems as the ecological success of these symbioses is not limited to coral reefs.

Introduction.

R1: General comment. The introduction provides the details necessary for the role of nutrient exchange in the symbiotic system, but lacks an equal attention on light intensity. Given the second experiment focuses on light, perhaps the introduction should provide some information on the current state of knowledge for light intensity on the symbiotic system.

A: Great suggestion. The introduction has been extended to include the following paragraph:

L57: "The photosynthetic activity of algal symbionts implies that the functioning of the cnidarian-algal symbiosis is intimately linked to light availability (7). Latitudinal, seasonal and tidal fluctuations in light intensity, attenuation with depth, shading, and turbidity differences create a complex mosaic of light conditions in aquatic environments (8–10). Such variability poses a challenge to photosynthetic organisms as low light levels may limit photosynthetic carbon fixation while high light levels may result in excessive photooxidative damage (7, 11). Yet, cnidarian-algal symbioses can be found across a wide range of light regimes ranging from intertidal to mesophotic environments (12, 13). The key to this broad ecological tolerance lies in the efficient photo-acclimation of both symbiotic partners. Specifically, changes in symbiont densities, host pigments, morphology, and behavior (e.g., locomotion) may modulate light microenvironments for algal symbionts within the host (14–18). Likewise, changes in photosynthetic pigments and antioxidant levels of the algae enable optimal light harvesting while avoiding excessive photodamage (18–20). However, holobiont responses to changes in light availability are highly species- and context-dependent and are often confounded by changes in other environmental parameters (e.g., along depths gradients) (15, 21). Our understanding of the regulatory processes shaping the ecological niche of cnidarian-algal symbioses is thus limited."

R1: L54- phrase "shine a light" could be replaced with "could elucidate the evolutionary..."

A: Implemented as suggested.

R1: General comment. Can the end of the introduction strengthen the rationale and motivation for the general audience; for example, besides the fact that the factors are poorly understood for nutrient regulation in cnidarian-algal symbioses, is there a larger motivation/impact for this ...e.g. its relevance to the environment and/or future expected conditions.

A: Thanks for raising this point. We have now elaborated on the justification for the study.

L85: "However, while the importance of nitrogen cycling in the cnidarian-algal symbiosis is widely accepted, the factors regulating nitrogen availability *in hospite* remain poorly understood (36, 37). Previous studies propose that algal symbiont nitrogen limitation arises from symbiont-symbiont as

well as host-symbiont competition for inorganic nitrogen (30, 34, 38–43). Deciphering the factors regulating this interplay of intra- and interspecific nitrogen competition could be key to understanding the functional regulation of the symbiosis. Here, we thus investigated the nutritional and environmental controls of nutrient cycling in the cnidarian-algal symbiosis to elucidate the processes shaping its ecological niche and tolerance limits in the Anthropocene.”

Results & Discussion

R1: L 87 -why have the authors not considered a multi label experiment of ^{13}C -pyruvate and $^{15}\text{NH}_4$? The nanoSIMS 50L can measure both up to 6 masses simultaneously (e.g. ^{13}C -, ^{12}C -, $^{12}\text{C}^{14}\text{N}$ -, $^{12}\text{C}^{15}\text{N}$ -). It seems like a shortcoming in the experimental design, but perhaps there is an explanation?

A: Please refer to the response above. ^{13}C measurements have now been included.

R1: L90. The use of “and/or” which is it-the algal symbionts have limited access and lack machinery OR have limited access or lack machinery? This detail seems rather crucial to the experimental design.

A: This was rephrased to clarify that Symbiodinicaeae lost the molecular machinery for pyruvate utilization.

R1: L92. What is the label percent of $^{15}\text{NH}_4$? This is important for the SIMS measures, and should be described here or in the materials and methods.

A: Notably, we used artificial seawater to exclude any background nutrient “contamination”. The purity of used isotope labels (≥ 98 atom %) has now been included for all isotopes in the Material and Methods.

R1: L107- Can the authors clarify how it is known that the photosynthetic hosts can utilize the additional carbon for their amino acid synthesis. Is this from previous works or the experiment described in the study?

A: While there are older works on this, we recommend the preprint by Cui *et al.* on this topic showing that labile carbon (in their study glucose) is converted into carbon backbones for amino acid synthesis: <https://doi.org/10.1101/2022.06.30.498212>

This study was cited throughout this manuscript and the reference is now included in this sentence as well.

R1: Fig 1 explain what APE is in the caption, add in the number of ROIs or cells analyzed-this detail should also be added to the materials and methods.

A: We apologize for the oversight. Where relevant, figure legends now mention the number of replicates involved and define what APE means.

R1: L138-140. Seems like this speculation/interpretation could use a reference? Or a comparison to another study?

A: Valid point. References for bleaching and/or lack of algal proliferation in the dark (Rouan *et al.* 2022 *Mol Ecol*, Jinkerson *et al.* 2022 *Curr Biol*) and photooxidative stress at high light (Richier *et al.* 2008 *JEMBE*, Levy *et al.* 2016 *Roy Soc B*) have now been added.

R1: General comment: With the data acquired in cell abundances and chl a content, is there any indication that the cells are altering their chl a content relative to their cell size? e.g. under high/low light algal cells can alter the pigment content for energy conservation or dissipation.

A: Great point. Indeed, these responses have been observed for some species, but the patterns strongly depend on species and environmental context. We have now included a new normalization of chlorophyll content per algal cell (see Figure S3A). As you can see, the chlorophyll a content of algal

cells was not affected by light availability suggesting that other mechanisms of photoacclimation were in place. This is now also highlighted in the Results & Discussion:

L232: “However, chlorophyll *a* content of individual algal cells remained unaffected by light availability suggesting that long-term photo acclimation was driven by other holobiont responses, e.g., changes in host pigmentation (LM, $F = 0.0$, $p = 0.995$; Fig. S3A). Nonetheless, chlorophyll *a* content in relation to host protein content increased by more than two-fold towards the symbiosis tolerance limits compared to intermediate light levels (LM, $F = 7.4$, $p = 0.004$; Fig. S3B).”

Materials and methods

R1: L 277-278. Rephrase, seems like host mortality is not necessary in the first part of the sentence.

A: The sentence was split for clarity.

R1: General comment. Fixation procedures could lead to a dilution of the enrichment, the authors could consider making a statement in the methods section to explain this. Several papers have reported this for bacteria, perhaps similar response is valid for the host and algal cells.

A: Point well taken. The following has been included in the methods section:

L386: “Notably, these enrichment values are likely an underestimation of the actual enrichment levels for these organisms as fixation, dehydration, and resin embedding during sample preparation extract and dilute soluble compounds from the sample matrix. However, any methodological bias arising from this is consistent across samples, NanoSIMS images hence allow for a robust assessment of relative enrichment values.”

R1: L309. It is probably better to state that the $^{15}\text{N}/^{14}\text{N}$ ratios were based on the $^{12}\text{C}^{15}\text{N}/^{12}\text{C}^{14}\text{N}$.

A: Implemented as suggested.

R1: L312. I think “ion images” is a better descriptor than “maps.”

A: Replaced as suggested.

R1: L329-330. Have the authors also tried to normalize to time in order to report rate?

A: We have now normalized bulk isotope measurements for experiment 2 per day and symbiont sell to allow easy comparison with oxygen measurements.

Figures

R1: General comment. Authors should consider to move and combine Fig S1 and Fig S2 to replace Fig. 3. These two supplementary figures are quite clear overview of the results from Exp. 2.

A: Great suggestion. Former Figures S1 and S2 have been moved to the main text to replace the previous figure.

R1: Suggestion/Question on analyses. Given that nanoSIMS is an amazing MSI technique with high lateral resolution, have the authors tried to draw and compare regions of high enrichment in the individual Symbiodinium. In Fig S2, a striking observation is the variation in the enrichment within the subpopulations of Symbiodinium in the sections-one can see clusters of cells of high enrichment co-occurring with cells of less enrichment. Furthermore, within a Symbiodinium smaller areas of extreme enrichment e.g., are these representative of storage granules? Is this variation due to variable physiology amongst the Symbiodinium OR related to the depth of the sectioning into the Symbiodinium? This is an important technical caveat in SIMS imaging on populations of cells embedded (or in colonies, etc), as they are not all in the same orientation, and hence one should comment/explain this variation. One could measure the diameter of the Symbiodinium, and plot the enrichment as a function of the cell diameter. This figure also makes me wonder how the ROIs for the host tissue were made? Perhaps the authors can show this as a Suppl. figure as well.

A: Indeed, NanoSIMS is an excellent tool to illustrate and analyze single-cell variability. However, in

the present datasets, algal cell ROI size had no effect on ^{15}N enrichments beyond introducing high variability in very small ROIs (which were hence excluded from the analysis where possible). We have now updated the methods accordingly and included a new Figure S5 showing that algal ROI size did not affect ^{15}N enrichment.

Further, we have included an additional Figure S4 to illustrate how ROIs correspond to SEM and NanoSIMS images and have extended the methods to include the following:

L375: “ ^{15}N assimilation was quantified by drawing regions of interest (ROIs) of host epidermis, host gastrodermis, and algal symbionts based on $^{12}\text{C}^{14}\text{N}^-$ images, respectively (**Fig. S4**). As individual host cells were not clearly distinguishable in the NanoSIMS images, all epidermal and gastrodermal tissue areas within one image (excluding algal symbionts and symbiosomal contents) were recorded as one ROI per tissue layer, respectively (**Fig. S4**). For experiment 1, this yielded 8 epidermal and gastrodermal ROIs per *Aiptasia* respectively. For experiment 2, this yielded 4 epidermal and 8 gastrodermal ROIs per *Aiptasia*. Algal symbionts ROIs were drawn based on individual algal cells and only the largest ROIs (40 ROIs per *Aiptasia* in experiment 1 and 16 ROIs per *Aiptasia* in experiment 2) were included in the analysis to minimize potential measurement variability due to lower signal-to-noise ratio of smaller ROIs. Algal symbiont ROI size thus had no significant effect on ^{15}N enrichments (LM, $F = 1.5$, $p = 0.224$ for experiment 1, $F = 0.1$, $p = 0.720$ for experiment 2; **Fig. S5**).”

Reviewer #2

R2: The cnidarian-phototroph symbiosis in principle relies on the transfer of photosynthetically-derived carbon from the symbiont to the host, while the host provides ideal conditions for the symbiont to thrive. It has been shown previously that increased nitrogen availability can lead to increased symbiont growth and subsequent decreases in the carbon transfer to the host (e.g. O’Neil and Capone, 2008) suggesting a tight coupling and/or host regulation of the symbiotic metabolism under stable environmental conditions. Here, Rädercker et al. present single-cell ammonium uptake by an anemone host (*Aiptasia*) and their photosynthetic symbionts under different carbon availabilities, accomplished by either additions of pyruvate or changes in light (and resulting changes in photosynthesis by the symbionts). The key finding is that when pyruvate is added to incubations of symbiotic *Aiptasia*, the ammonium uptake of the host increases while the ammonium uptake of the symbiont decreases (Fig. 1G). The authors argue that under these increased external (organic) carbon availabilities (i.e. the pyruvate addition), the host competes for ammonium with its symbionts, leading to the decreased ammonium uptake by the symbionts. This conclusion would indicate that the host could, to some extent, control the availability of nitrogen to the symbionts and the subsequent transfer of carbon to themselves.

I see one major issue with the authors’ argumentation. While I do not necessarily disagree that the host may have the capacity to control symbiont growth (via coupling of carbon and nitrogen cycling), the authors do not provide enough evidence of the ‘resource competition’ between host and symbiont or that nitrogen availability was different for host and symbiont. In more detail, the authors used only one concentration of ammonium in their experiments, i.e. ten micromolar ammonium, and do not show that i) this concentration was limiting during the incubation or ii) that ammonium was used up during the incubation. Generally, competition for a resource, such as nitrogen, should only occur if substrate availability is limiting. However, the authors do not provide evidence that this is the case here. In contrast, the ten micromolar ammonium concentration is substantially more than typical in situ concentrations in the oligotrophic conditions of coral reef ecosystems. In addition, incubations using ¹³C-labeled compounds could have possibly provided the means to constrain the coupled carbon-nitrogen cycling but such data is not provided here. These types of incubations/approaches could have provided additional information on what type of carbon and nitrogen cycling was occurring, and how strongly carbon and nitrogen cycling were coupled. In summary, this manuscript does not provide enough evidence for the authors’ conclusions.

A: Thank you for this very constructive feedback and the interest in our work.

Indeed, ammonium concentrations used here exceed concentrations found on most reefs. Yet, these concentrations are necessary to ensure sufficient signal for a robust quantification using NanoSIMS imaging and to account for the depletion of ammonium over time. As suggested, we have now included additional measurements to show that the uptake of ammonium from the seawater is limited by carbon (pyruvate) availability and that ammonium can be depleted to the limit of detection within 3 hours of incubation (Figure S1). Hence, nitrogen availability was indeed limited for the holobiont supporting the interpretation that host and symbionts compete for this valuable resource.

Further, we have now included additional ¹³C stable isotope measurements (based on bulk analysis), which show that 1) only the host assimilates pyruvate effectively and 2) photosynthetic carbon assimilation is highest at intermediate light levels within the cnidarian-algal symbiosis.

We are thus confident that the reviewer’s suggestion to include these additional measurements substantiated our previous interpretations and has strengthened the narrative of the manuscript.

Among other sections, the results & discussion now include the following paragraph:

L107: “Previous studies suggest that algal symbionts may lack the cellular machinery to utilize pyruvate (48, 49). Indeed, bulk isotope analysis confirmed that host tissues of aposymbiotic and symbiotic *Aiptasia* efficiently assimilated [2,3-¹³C]-pyruvate, while ¹³C enrichments of algal symbionts were two orders of magnitude lower (Fig. S1A). Pyruvate addition hence increased labile carbon

availability for the host but not the algal symbiont, allowing us to study how the availability of labile carbon in the host metabolism affects ammonium assimilation by symbiotic partners. Consistent with previous observations (50), incubations with 10 μ M ammonium revealed that aposymbiotic *Aiptasia* showed net release of ammonium during 6 h incubations. In contrast, symbiotic *Aiptasia* showed net uptake of ammonium from the seawater (Tukey HSD, $p < 0.001$). Pyruvate addition, however, increased ammonium uptake by aposymbiotic and symbiotic *Aiptasia* (ANOVA, $F = 181.8$, $p < 0.001$), resulting in a depletion of ammonium in the seawater to the limit of detection within 3 h of incubation (Fig. S1B-D). The retention and uptake of ammonium by the *Aiptasia* holobiont thus appear to be limited by labile carbon availability. Algal photosynthesis or environmental carbon sources (here pyruvate) thus enhance metabolic nitrogen demand in the holobiont, likely resulting in nitrogen limitation for symbiotic partners under oligotrophic conditions.”

R2: 37: Please clarify what is meant by ‘enhanced symbiotic competition’

A: Rephrased to clarify that increased host nitrogen assimilation is limiting nitrogen availability for the algae.

R2: 39: Please clarify what is meant by ‘tolerance limits of the symbiosis’. While this is explained later in the manuscript, I think it is essential that the reader already understands in the abstract what is meant.

A: The abstract now states that this refers to the lower and upper light tolerance limits of the stable symbiosis.

R2: 83/86: I don’t think that ammonium is ‘fixed’; rather it is taken up and assimilated or incorporated. Using the term ‘fixing’ or ‘fixation’ may mislead the reader to think that N₂ fixation plays a role here.

A: Changed to “assimilation” throughout the manuscript.

R2: 107: How do you know whether the pyruvate was used for amino acid synthesis and not for, for example, lipid biosynthesis?

A: Importantly, we do not claim that pyruvate is exclusively used for amino acid synthesis and nothing else. Indeed, it is very likely, that excess carbon will also be converted into lipids and other forms of carbon storage when nitrogen is limited. However, our data clearly show that increased labile carbon availability stimulates ammonium assimilation in the animal as a consequence of amino acid synthesis. The sentence in question has been rephrased for clarity:

L139: “...even symbiotic hosts were able to utilize the increased labile carbon availability to stimulate their amino acid synthesis...”

R2: 176: Doesn’t the light mediate photosynthesis and therefore carbon rather than N availability?

A: Indeed. The link between photosynthetic carbon and nitrogen availability is highlighted in the previous sentence. The sentence in question has been reworded to clarify that this statement refers to the previous results:

L228: “Given that nitrogen limitation controls algal growth in the cnidarian-algal symbiosis (30, 31, 36, 37), the here-observed light-dependent changes in symbiotic nitrogen availability should directly affect the regulation of symbiosis.”

R2: Fig. 1: It would be good to have some microscopic context to the nanoSIMS images (and/or outlines of tissues and so on).

A: Great suggestion. We have now included a correlated SEM & NanoSIMS image as part of Figure S4 to help the reader understand the NanoSIMS images and have extended Figure 1 to include a label indicating the position of the mesoglea.

R2: Fig. 1: The panels H and I in Fig. 1 do not seem to show substantial differences in the ¹⁵N-enrichment in the symbionts between the control and +pyruvate treatment. They do not seem to reflect the findings shown in panel G.

A: Given that host and symbiont enrichment differ by nearly one order of magnitude, the selected color scale reflects a compromise between illustrating changes in the host and the symbiont tissue. Even in this color scale, algae in panel H are homogeneously pink (maximum enrichment) while algae in panel I show larger areas of blue (no enrichment) or yellow (intermediate enrichment).

R2: Fig. 1: In panel C, I would rather show the 'Epi' and 'Gas' data on top of the 'Epi' and 'Gas' data in label G, highlighting that there are no symbionts in the aposymbiotic Aiptasia.

A: Implemented as suggested.

References:

R2: O'Neil, Judith M., and Douglas G. Capone. "Nitrogen cycling in coral reef environments." *Nitrogen in the marine environment* (2008): 949-989.

A: This study is now cited in the manuscript.

Reviewer #3

R3: In this manuscript, Raedecker and colleagues investigate how organic C availability shapes competition for nitrogen and regulates symbiont abundance in the model photosymbiotic system *Aiptasia* (*Exaiptasia diaphana*) and its phototrophic *Symbiodinium* symbionts. The authors compare ¹⁵N-ammonium assimilation under labile C-replete vs. C-deplete conditions, in symbiotic and aposymbiotic *Aiptasia* on single cell level using nanoSIMS. Furthermore, they track symbiont abundance and activity under varying light regimes, leading to differing availabilities of organic C from photosynthates for the host. This manuscript sheds light on the mechanism sustaining (and destabilizing) photosymbiotic systems, and adds to the growing body of literature highlighting nitrogen availability as key factor for this type of symbiosis. While I am not an expert in photosymbiosis, I very much enjoyed reading this interesting and well written manuscript and particularly liked the elegant experiments and combination of methods from single cell to bulk activity measurements. I do have some comments on the manuscript, mainly pertaining to the methodological setup (e.g. the rather high concentration of organic C used in this study), as detailed below.

A: Thank you for the encouraging feedback. We appreciate your interest in our work. Please see below for a detailed response to your comments.

Main points:

R3: - I would like to hear the authors' opinion on the transferability of the labile C experiments to in situ conditions. The used labile C concentration (10 mM pyruvate) seems rather extreme compared to the natural conditions these animals live in (typical surface seawater DOC concentration is 100-500 μM). I wonder whether the highly elevated pyruvate concentrations used here might distort the host (and symbiont) responses in ¹⁵N-assimilation reported in the manuscript. Was the response to lower (more natural) pyruvate additions tested?

A: Importantly, the purpose of the pyruvate treatment was not to mimic natural conditions. Rather, we wanted to use this carbon source to selectively enhance carbon availability for the host but not the algal symbiont. As such, we chose a high concentration to ensure sufficient uptake by the host. Even at the used concentrations, ¹³C signature of pyruvate-labeled animals were still below those of bicarbonate-labeled animals from experiment 2 (even when considering the longer incubation time in experiment 2). Hence, we are confident that pyruvate addition is a powerful experiment to test the effect of photosynthate translocation on the host metabolism and its consequences for symbiotic nitrogen competition.

We now elaborate further on the reasoning behind choosing pyruvate labeling in the results & discussion:

L107: "Previous studies suggest that algal symbionts may lack the cellular machinery to utilize pyruvate (48, 49). Indeed, bulk isotope analysis confirmed that host tissues of aposymbiotic and symbiotic *Aiptasia* efficiently assimilated [2,3-¹³C]-pyruvate, while ¹³C enrichments of algal symbionts were two orders of magnitude lower (**Fig. S1A**). Pyruvate addition hence increased labile carbon availability for the host but not the algal symbiont, allowing us to study how the availability of labile carbon in the host metabolism affects ammonium assimilation by symbiotic partners."

R3: To me it is unclear how many replicates were measured on the nanoSIMS. Please state this in the figure legend (fig 1) and in the main text. This is important to know, as it can otherwise not be assessed how much of the observed differences between +/- pyruvate is due to biological heterogeneity of individual animals, or response to the treatment.

A: Thanks for this suggestion. All figure legends now state the number of replicates used in the respective analysis.

R3: Likewise, I would suggest to show data from individual *Aiptasia* as separate points in Fig 2, 3, and the supplementary figures, to show variability between individual animals.

A: We agree that it would be desirable to plot all raw data where possible. We have now included plots showing all raw data associated with both experiments as part of the Zenodo submission. For experiment 1, we have included the raw data in the individual figures of the manuscript. For experiment 2, however, we have chosen to stick with plotting mean \pm se in the main manuscript for two reasons: 1) Some of the figures appear too crowded when all individual responses are included. 2) The spread of the raw data results in an extended y-axis making the shape of the response curve less visible to the reader in some cases.

Minor comments:

Introduction:

R3: I suggest to briefly introduce the photosymbiotic system (species) used here, and especially mention the Symbiodiniaceae symbionts by name.

A: The introduction has been extended to include:

L91: “*Exaiptasia diaphana* clonal line CC7 harboring *Symbiodinium linucheae* strain SSA01 endosymbionts”

Results & Discussion:

R3: L100: Do the authors have an explanation for the lack of significant response to pyruvate addition in the gastrodermal tissue of the photosymbiotic animals? Higher general labile C-availability due to the physical proximity to the symbionts and thus lack of response to elevated external C?

A: We believe there is a misunderstanding here, L100 stated that the gastrodermis showed a significant increase (note the log scale in the figure, this reflects an 80 % increase between treatments). L99 however, states that no significant increase was observed for the epidermis. Unfortunately, we can only speculate regarding the absence of a response in the epidermis. Potentially, pyruvate is more efficiently assimilated in the gastrodermal tissue of symbiotic animals or the lower abundance of ammonium transporters throughout the epidermis limits higher ammonium assimilation. The latter hypothesis is now mentioned in the manuscript:

L129: “However, this effect was restricted to the gastrodermal tissue, i.e., the tissue hosting the algal symbionts (Tukey HSD, $p = 0.979$ for the epidermis and $p < 0.001$ for the gastrodermis), potentially reflecting the elevated expression and localization of bi-directional ammonium transporters reported for gastrodermal cells in symbiotic *Aiptasia* (43).”

R3: Can the authors make a rough calculation on how much of the added 15N-ammonium was consumed by the holobiont in the 6h (pyruvate experiment) and 24h (light levels experiment)? Was there residual ammonium available?

A: Great suggestion. In line with the other reviewers’ feedback, we have now included additional ammonium uptake experiments to show that pyruvate-supplemented animals are able to deplete ammonium in the incubation vial within 3 h of incubation. Even symbiotic animals were able to deplete ammonium concentrations by about 70 % within 6 h of incubation suggesting that 24 h incubations are long enough for animals to deplete ammonium concentrations under optimal light conditions. Hence, both experiments presented here allow us to study the partitioning of ammonium under nitrogen-limited conditions. These additional measurements are now presented in Figure S3 and the Results & Discussion now include the following:

L112: “Consistent with previous observations (50), incubations with 10 μ M ammonium revealed that aposymbiotic *Aiptasia* showed net release of ammonium during 6 h incubations. In contrast, symbiotic *Aiptasia* showed net uptake of ammonium from the seawater (Tukey HSD, $p < 0.001$). Pyruvate addition, however, increased ammonium uptake by aposymbiotic and symbiotic *Aiptasia* (ANOVA, $F = 181.8$, $p < 0.001$), resulting in a depletion of ammonium in the seawater to the limit of detection within 3 h of incubation (**Fig. S1B-D**). The retention and uptake of ammonium by the *Aiptasia* holobiont thus appear to be limited by labile carbon availability. Algal photosynthesis or environmental carbon

sources (here pyruvate) thus enhance metabolic nitrogen demand in the holobiont, likely resulting in nitrogen limitation for symbiotic partners under oligotrophic conditions.”

R3: Figure 2: In the legend, descriptions of panel A and B are swapped. Please explain the abbreviation APE in the legend (atom percent excess), this is also missing in the other figures depicting nanoSIMS or isotope enrichment data. Please double check the data sheet for the nanoSIMS data of experiment 2 – AP and APE columns contain identical values.

A: Thanks for spotting this important point. All figure legends have been corrected and now include number of replicates as well as definitions of abbreviations. Further, the supplementary data files have been updated to now include the correct AP and APE values of ¹⁵N assimilation.

Methods:

R3: L313-316: I don’t understand what is meant by this statement. Please specify how ROIs of the host were drawn. The data sheet suggests that the areas of host ROIs were much larger than the symbiont ROIs – I find this surprising, as I would have intuitively drawn ROIs around individual host cells, which should not be that much larger than the symbionts, to depict the variability in host tissue enrichment.

A: We agree that it would be ideal to analyze individual host cells as separate ROIs. However, neither SEM nor NanoSIMS images provided sufficient information to allow identifying individual host cells. As any segmentation of host tissues would thus be arbitrary, we decided to group host tissue area in only one host epidermal and gastrodermal ROI per image. To clarify, how ROIs were drawn we have now included supplementary Figure S1. Further the methods now state:

L376: “As individual host cells were not clearly distinguishable in the NanoSIMS images, all epidermal and gastrodermal tissue areas within one image (excluding algal symbionts and symbiosomal contents) were recorded as one ROI per tissue layer, respectively (Fig. S4). For experiment 1, this yielded 8 epidermal and gastrodermal ROIs per Aiptasia respectively. For experiment 2, this yielded 4 epidermal and 8 gastrodermal ROIs per Aiptasia. Algal symbionts ROIs were drawn based on individual algal cells and only the largest ROIs (40 ROIs per Aiptasia in experiment 1 and 16 ROIs per Aiptasia in experiment 2) were included in the analysis to minimize potential measurement variability due to lower signal-to-noise ratio of smaller ROIs.”

R3: Bulk elemental analysis: Please specify how Aiptasia individuals were washed before elemental analysis (I assume they were washed, else, ¹⁵N ammonium likely sticks to the biomass and biases the measurement)

A: Thanks for spotting this. The methods now clarify that all samples were rinsed in artificial seawater without isotope tracers before the samples were prepared for analysis.

R3: L347-352: Is the respiration rate of the host identical in light and dark conditions? This would need to be the case to be able to calculate gross photosynthesis the way it was done. This could be at least roughly assessed by measuring oxygen consumption by aposymbiotic Aiptasia in light vs. dark conditions.

A: The here-used method is the most commonly employed technique to measure gross photosynthesis in cnidarians (e.g., Ferrier-Pagès et al. 2000 Coral Reefs, Cardini et al. 2015 Roy Soc B, Carbonne et al. 2021 Limnol Ocean). Nonetheless, we acknowledge that this method cannot account for potential increases in light respiration rates due to enhanced carbon availability. As such, the reported gross photosynthesis may underestimate actual rates. However, any methodological bias arising from this is consistent across treatments and thus doesn’t affect the conclusions presented here.

This methodological bias is now acknowledged in the methods:

L434: “Notably, this method does not account for potential increases in respiration rates during the light (65). The here presented gross photosynthesis rates thus likely represent an underestimation of

the actual rates. However, this methodological bias is consistent across treatments and does not impair the conclusions presented here.”

R3: L354-374: Can the authors confirm that no (or very few) symbiont cells are retained in the supernatant after pelleting?

A: Indeed, we quantified the symbiont content in the host supernatant as well to confirm that symbionts were efficiently pelleted by centrifugation in all samples. This information is not included in the manuscript.

L397: “...host and algal symbiont fractions were separated by centrifugation (1000 g for 5 min, sufficient to pellet > 95 % of algal symbionts from the sample)...”

REVIEWERS' COMMENTS

Reviewer #1 (Remarks to the Author):

All of my considerations and suggestions were properly and thoroughly addressed. I appreciate the detailed and thoughtful responses to all my comments. The additional figure (S4) and details in the methods and nanoSIMS analyses are important improvements and well implemented. THANK you for the opportunity to review this important and timely work.

Reviewer #2 (Remarks to the Author):

Review of revised manuscript:

I am very happy to see that the authors were able to provide additional data to support their conclusions. Particularly, the addition of the ammonium concentrations is very valuable to show that indeed N limitation sets in after some hours and that therefore organisms are likely competing for it. All other comments have also been resolved, and the manuscript is really well written and structured. I have no further comments.

Reviewer #3 (Remarks to the Author):

Thank you for addressing my previous comments on the manuscript.

The additional data on ¹³C-pyruvate uptake by host and symbionts, and the inclusion of the ammonium concentration data have further strengthened the manuscript.

I have only few remaining suggestions/comments:

L89-90: Ecological niche and tolerance limits of this symbiosis were probably the same/similar in their recent evolutionary history also before entering the Anthropocene - please rephrase.

L116: maybe better "...ammonium uptake by *both* aposymbiotic and symbiotic..." for clarity

L134-135: Is it possible that the 10mM pyruvate addition has an inhibitory/detrimental effect on the symbionts' activity? Has this been tested in symbiont pure culture?

L141, L310: please clarify what starvation refers to here - I assume no feeding with Artemia, but it could also mean no light.

L206-209: Can higher ^{15}N enrichment by the symbionts at lower light intensities not simply be due to less excess released organic C from the symbionts due to lower light stress, and thus less available organic C for the host (and thus less host N-assimilation), and a more tightly coupled C fixation : N assimilation ratio by the symbionts?

L278: While the experiments conducted look at light intensities, the discussion about environmental conditions is about heat stress - it might be useful for readers to link this more explicitly in the discussion.

L323: The methods section states that 3 animals were sampled for nanoSIMS, the nanoSIMS figure legends state that 2 were analyzed. Please mention in the methods section that two replicates were analyzed by nanoSIMS.

Response to reviewer comments
Coupled carbon and nitrogen cycling regulates the cnidarian-algal symbiosis

Comments by the reviewers are indicated in **black**.

Responses by the authors are indicated in **blue**.

New and revised manuscript sections are indicated in **green**.

Reviewer 1:

R1: All of my considerations and suggestions were properly and thoroughly addressed. I appreciate the detailed and thoughtful responses to all my comments. The additional figure (S4) and details in the methods and nanoSIMS analyses are important improvements and well implemented. THANK you for the opportunity to review this important and timely work.

A: We thank the reviewer for their constructive feedback and support.

Reviewer 2:

R2: Review of revised manuscript:

I am very happy to see that the authors were able to provide additional data to support their conclusions. Particularly, the addition of the ammonium concentrations is very valuable to show that indeed N limitation sets in after some hours and that therefore organisms are likely competing for it. All other comments have also been resolved, and the manuscript is really well written and structured. I have no further comments.

A: We thank the reviewer for their constructive feedback and support.

Reviewer 3:

R3: Thank you for addressing my previous comments on the manuscript.

The additional data on ¹³C-pyruvate uptake by host and symbionts, and the inclusion of the ammonium concentration data have further strengthened the manuscript.

A: We thank the reviewer for their constructive feedback and support. Please see below for detailed responses.

R3: I have only few remaining suggestions/comments:

L89-90: Ecological niche and tolerance limits of this symbiosis were probably the same/similar in their recent evolutionary history also before entering the Anthropocene - please rephrase.

A: Agreed. The reference to the Anthropocene was removed from the sentence accordingly.

R3: L116: maybe better "...ammonium uptake by *both* aposymbiotic and symbiotic..." for clarity

A: Implemented as suggested.

R3: L134-135: Is it possible that the 10mM pyruvate addition has an inhibitory/detrimental effect on the symbionts' activity? Has this been tested in symbiont pure culture?

A: While we did not test the effects of pyruvate on algal cultures, we did test prolonged pyruvate exposure in *Aiptasia*. Over the course of two-week experiments, no decline in algal photosynthetic efficiency or pigmentation was observed. Hence, it is unlikely that the used pyruvate concentrations had severe negative effects on algal symbionts or their host.

R3: L141, L310: please clarify what starvation refers to here - I assume no feeding with *Artemia*, but it could also mean no light.

A: Indeed. We now clarify: "...without heterotrophic nutrient sources..."

R3: L206-209: Can higher ¹⁵N enrichment by the symbionts at lower light intensities not simply be due to less excess released organic C from the symbionts due to lower light stress, and thus less

available organic C for the host (and thus less host N-assimilation), and a more tightly coupled C fixation : N assimilation ratio by the symbionts?

A: Light stress is unlikely a major driver of carbon translocation in the symbiosis. If high algal ¹⁵N enrichment would indicate reduced competition by the host, host and algae should have inverted ¹⁵N enrichment patterns. Yet, both symbiotic partners show their highest ¹⁵N enrichment at intermediate light levels. Hence, we propose that increased nitrogen competition reduces host catabolic nitrogen release and increases the relative proportion of environmental ammonium in the nitrogen pool of either symbiotic partner. For clarification, we have rephrased this section:

“While ¹⁵N enrichments only reflect the uptake of environmental nitrogen in the symbiosis, the recycling of ammonium from the host catabolism is likely regulated by the same processes and follows the same patterns. High ¹⁵N enrichments thus indicate an increased nitrogen demand in the symbiosis resulting in reduced production of catabolic ammonium by the host.”

R3: L278: While the experiments conducted look at light intensities, the discussion about environmental conditions is about heat stress - it might be useful for readers to link this more explicitly in the discussion.

A: Great suggestion. The introduction of the paragraph has been rephrased for clarity:

“Finally, the processes described here improve our understanding of the maintenance of the symbiosis in a changing environment. These processes are unlikely limited to the light-dependent regulation of the symbiosis but likely control other acclimation and stress responses as well. Specifically, our findings may contribute to deciphering the processes leading to the breakdown of cnidarian-algal symbiosis during heat stress (54).”

R3: L323: The methods section states that 3 animals were sampled for nanoSIMS, the nanoSIMS figure legends state that 2 were analyzed. Please mention in the methods section that two replicates were analyzed by nanoSIMS.

A: Thanks for spotting! In this case, the figure legend was wrong. Indeed, 3 animals were analyzed per treatment for experiment 1. The figure legend for Fig. 1 has been updated accordingly.